# Silicon Photocatalytic Water-Treatment: Synthesis, Modifications, and Machine Learning Insights

**DOI:** 10.3390/nano15191514

**Published:** 2025-10-03

**Authors:** Abay S. Serikkanov, Nurlan B. Bakranov, Tunyk K. Idrissova, Dina I. Bakranova, Danil W. Boukhvalov

**Affiliations:** 1National Academy of Sciences of the Republic of Kazakhstan Under the President of the Republic of Kazakhstan, Almaty 050000, Kazakhstan; 2Institute of Physics and Technology, Satbayev University, Almaty 050000, Kazakhstan; 3Research Group altAir Nanolab LLP, Almaty 050000, Kazakhstan; 4Firma Balausa LLP, Almaty, Kazakhstan; 5Faculty of Engineering & Natural Sciences, SDU University, Kaskelen 040900, Kazakhstan; 6Institute of Materials Physics and Chemistry, College of Science, Nanjing Forestry University, Nanjing 210037, China

**Keywords:** Si-based nanostructures, photocatalytic water treatment, heterostructures, silicon doping, machine learning

## Abstract

Photocatalytic technologies based on silicon (Si-based) nanostructures offer a promising solution for water purification, hydrogen generation, and the conversion of CO_2_ into useful chemical compounds. This review systematizes the diversity of modern approaches to the synthesis and modification of Si-based photocatalysts, including chemical deposition, metal-associated etching, hydrothermal methods, and atomic layer deposition. Heterostructures, plasmonic effects, and co-catalysts that enhance photocatalytic activity are considered. Particular attention is drawn to the silicon doping of semiconductors, such as TiO_2_ and ZnO, to enhance their optical and electronic properties. The formation of heterostructures and the evaluation of their efficiency were discussed. Despite the high biocompatibility and availability of silicon, its photocorrosion and limited stability require the development of protective coatings and morphology optimization. The application of machine learning for predicting redox potentials and optimizing photocatalyst synthesis could offer new opportunities for increasing their efficiency. The review highlights the potential of Si-based materials for sustainable technologies and provides a roadmap for further research.

## 1. Introduction

The growth of industry and the intensive use of hydrocarbon energy sources are pushing the planet’s ecosystems to the brink of crisis. Waste emitted into the atmosphere and waterways causes severe pollution, affecting approximately 1.8 billion people. Hence, some practical tools are needed to convert harmful compounds into safe ones. One promising method of environmental purification is the use of photocatalytic processes to remove hazardous organic molecules and microorganisms from water resources [1]. In addition to purification, photocatalytic technologies show promise in converting organic waste into electricity [2]. Photocatalysts based on semiconductor materials that are active under light are of broad interest [3,4]. In general, photocatalytic water treatment is based on the generation of reactive oxygen species, including hydroxyl radicals (OH), superoxide radical anions (O_2_^–^), and hydrogen peroxide (H_2_O_2_) [5]. Photocatalytic reactions destroy the cell walls of microorganisms, oxidize biomolecules, and result in the inactivation of bacteria, viruses, and other pathogens [6,7]. Photodegradation started when a semiconductor absorbs photons with energies equal to or larger than the bandgap width, resulting in the excitation of electrons (e^–^) from the valence band to the conduction band and the formation of holes (h^+^). The valence and conductive bands’ potentials are essential for this process [8]. The excited charges can recombine or migrate to the photocatalyst surface, where they react with adsorbed organic molecules [9,10]. Figure 1 shows a scheme of the mechanism of photocatalytic degradation of organic dyes using nanomaterials under the influence of radiation, where the photocatalyst divided by a dotted line into two zones: the upper zone corresponds to the conducting zone with an electron (e^–^_CB_), the lower zone corresponds to the valence zone with a hole (h^+^_VB_), which illustrates the charge separation under photoinduced excitation. The adsorbed oxygen (O_2 ads_) interacts with the electron of the conducting band (e^–^_CB_) to form superoxide anion (O_2_^–^). The superoxide anion is then converted to hydrogen peroxide (H_2_O_2_^–^) in the presence of protons (H^+^). Hydrogen peroxide can additionally react with an electron (e^–^_CB_), generating hydroxyl radicals (OH^–^) and hydroxide ions (OH^–^). In parallel, the valence band hole (h^+^_VB_) oxidizes water molecules (H_2_O), also producing hydroxyl radicals (OH^–^) and protons (H^+^), or interacts with H_2_O_2_, enhancing radical formation [11].

Silicon (Si), due to its availability, low cost, tunable bandgap, and high biocompatibility, is considered a promising material for photocatalytic applications, including water purification, hydrogen generation, reduction of CO_2_ and NO to small-molecule fuels and chemicals, and fixation of CO_2_ in organic compounds to obtain valuable substances [12,13]. Nevertheless, despite the high potential for photocatalysis and optoelectronics, the stability of Si-based structures is limited due to oxygen sensitivity [14]. The oxidation of the silicon surface reduces stability and necessitates the use of sacrificial reagents to trap holes, thereby compromising the environmental friendliness of the process [15]. To improve the stability and efficiency of photocatalytic Si-based structures, systems are being developed that utilize surface plasmon resonance and employ nanoengineering and co-catalysts to enhance the rate of photocatalytic reactions [16]. In addition to its direct purpose for photocatalytic reactions, silicon is also widely used as a substrate for the deposition of advanced photoactive surfaces [17]. Hydrothermal techniques for depositing highly crystalline photocatalysts on silicon substrates have gained widespread use in recent years [18,19].

This mini-review discusses the photocatalytic properties of silicon, methods of synthesis of efficient Si-based photocatalysts, heterostructures, plasmonic, and co-catalysts. The review aims to systematize current approaches and recent achievements in the development of Si-based photocatalysts, to analyze their advantages and limitations, and to assess the prospects of applying machine learning for their optimization. Abundance and low cost still keep silicon a promising material for photocatalysis despite the appearance of multiple novel compounds. However, its efficiency and stability depend on the synthesis and modification methods, which are discussed below.

## 2. Synthesis and Modification of Si-Based Nanostructures

Si-based nanostructures are synthesized and modified using a variety of methods, which are classified into bottom-up and top-down approaches. These approaches enable the creation of nanostructures with unique properties, including a high surface area, improved optical and electronic characteristics, and enhanced mechanical robustness [20]. Among the many methods, chemical deposition, metal-assisted chemical etching (MACE), hydrothermal methods, different types of reduction (carbothermal, magnesiothermal, electrochemical), atomic layer deposition (ALD), electrodeposition, mechanical approaches (doping, impact pressing), as well as thermal evaporation, sol-gel methods, template synthesis, focused ion beam (FIB), and laser ablation can be mentioned. Each method has specific advantages and limitations that determine its suitability for creating nanostructures with desired characteristics.

Chemical deposition allows the creation of arrays of silicon pyramids (SiPYs) decorated with PbS nanoparticles on a Si substrate. This combination of surface shape and doping yields a synergistic effect through efficient charge transfer, thereby enhancing the activity in the reduction of CO_2_ to low-molecular-weight fuels [21]. The method is characterized by its simplicity and the availability of equipment, allowing for the control of the shape and size of nanostructures. However, it is limited in forming complex structures, and its sensitivity to reaction conditions reduces the reproducibility of the results. For example, the nanoporous Si obtained by this method exhibits a quantum yield (AQY) of 12.1% at 400 nm, due to its crystalline structure [22]. This result highlights the importance of morphology in photocatalysis. The challenge of precisely controlling reaction parameters makes the process difficult to scale up and implement practically, underscoring the need for further research.

Metal-assisted chemical etching (MACE) is used to create silicon nanowires (SiNWs) on p-type Si(100) and Si(111) monocrystalline substrates with a resistivity of 0.009–0.010 Ω·cm in an HF/H_2_O_2_ solution. The orientation of the substrate determines the morphology of the nanowires. On Si(100), two-bond atoms facilitate vertical growth, whereas on Si(111), three bonds lead to anisotropic etching, resulting in inclined or zigzag structures. The concentration of HF also plays a key role: low values promote the formation of Si oxide, which slows down the process and forms ordered NWRs. The high values equalize oxidation and dissolution, thereby reducing the degree of order. The process includes degreasing in acetone and isopropanol, washing with Milli-Q water, cleaning in pyranium solution (H_2_SO_4_/O_2_) at 80 °C, etching in a mixture of HF (5 M) and AgNO_3_ (0.035 M) at 55 °C for 30 min, removal of Ag by washing in HNO_3_ and drying in a nitrogen stream [23,24]. MACE offers high morphology control and produces structures with a large surface area, making it ideal for photocatalysis. However, the dependence on substrate orientation limits the versatility of the method by requiring specific types of Si wafers, and the need for precise control of reagent concentrations and multi-step purification complicates standardization and increases operating costs [25]. Usage of strong acids also limits environmental friendliness of this technique after scaling up.

Hydrothermal methods utilize high temperatures and pressures in an aqueous environment to deposit highly crystalline, photoactive coatings on Si substrates [18,19,20]. They provide excellent adhesion and crystallinity, which improves photocatalytic activity. However, the need for autoclaves and high energy consumption make the process expensive and less suitable for mass production, although the precision of the method is justified for high-performance niche applications [26]. Therefore, the hydrothermal methods are less attractive for large-scale synthesis, although their precision is justified for high-throughput applications.

The carbothermal reduction of SiO_2_ to pure Si using carbon at high temperatures is an economical and widely used industrial process [27]. The method is suitable for creating SiC-SiO_2_ nanotubes [28]. However, high temperatures increase energy consumption, and the formation of by-products, such as SiC, requires additional purification. Magnesiothermal reduction at 600–700 °C reduces the temperature requirements, but is also accompanied by the formation of SiC, which reduces product purity. Reduction of molten salts at 200–260 °C yields approximately 40% and utilizes low temperatures. On the other hand, the use of aggressive reagents, such as molten salts, limits scalability [29].

Electrochemical reduction at temperatures below 850 °C enables the process to be controlled through electrical potential. The use of reagents such as CaO and Na_2_O can damage the equipment and make this technology environmentally unfriendly [29]. Although these reduction methods are economical for mass production, their disadvantages, such as the formation of by-products and low yields, indicate the need to develop more efficient processes [30,31,32,33].

Atomic layer deposition (ALD) is used to fabricate complex heterostructures such as Pd/TiO_2_/Si nanopillars, providing highly precise and uniform coatings [34]. Despite recent progress, the high hardware cost and long processing time significantly limit scalability [35,36,37,38]. Electrodeposition applied to deposit Cu_2_O on CNT/Si forms suspended architectures with controlled morphology, but ensuring uniformity is challenging [39]. Mechanical alloying (ball-milling) and impact pressing synthesize Ti-Si composites with high mechanical strength. High kinetic energy enables the creation of nanostructured materials. The limited control over particle size is the key disadvantage of this technique [40].

Additional methods such as thermal evaporation, sol-gel synthesis, and FIB are described in recent works [41,42,43]. Highly crystalline samples (mp-Si300) synthesized at 300 °C had an ordered structure and a minimal number of defects, which ensured effective charge separation and high photocatalytic activity (4437 μmol H_2_·h^–1^·g^–1^). At the same time, low-crystalline samples (mp-Si100) obtained at 100 °C exhibited an amorphous structure with a large number of defects, which reduced charge transfer efficiency and increased recombination losses, limiting their photocatalytic activity [44]. The high activity of mp-Si300 highlights the role of crystallinity, but its influence on other morphologies, such as nanosheets or nanopores [45], requires further investigation. In addition to crystallinity, the oxidation rate of the active layers obtained requires attention. Atomically thin silicon nanosheets (A-SiNS) exhibit enhanced activity in photocatalytic reactions due to their high specific surface area and efficient charge migration. Studies show that the photocatalytic hydrogen production rate for A-SiNS is 158.8 mmol·h^–1^·g^–1^, which is 3.2 times higher than that of multilayer silicon nanosheets (M-SiNS) and significantly exceeds that of conventional silicon. However, due to surface oxidation, activity degradation is observed after three cycles of operation, which reduces the number of active centers and increases charge carrier losses, which makes this approach difficult for industrial applications [45]. Si NWs demonstrate high photocatalytic activity due to their increased surface area and improved charge transfer [46].

Reducing the size of silicon particles increases photocatalytic activity. Still, it is accompanied by a decrease in hydrogen yield due to the high density of surface states, which act as recombination centers for charge carriers. These works also show that low-purity metallurgical silicon exhibits unexpectedly high photocatalytic properties when its particle size is reduced. The rate of hydrogen evolution increases hundreds of times, but simultaneously, a significant degradation of photocatalytic activity is observed. The primary reason for this is the Mott-Schottky effect at the metal-silicon interface, which results in enhanced recombination of charge carriers. Improving the metal-silicon contact by forming a heterostructure is a promising solution for suppressing degradation and maintaining high photocatalytic efficiency [47].

The broad applicability of Si-based nanostructures is due to their availability, tunable electronic properties, and ability to form heterostructures, which opens up prospects for PC optimization. Silicon nanowires (SiNWs) synthesized by non-contact electrochemical etching using Ag [48] exhibit enhanced photoelectrochemical properties and can serve as active elements in solar energy conversion systems. Additionally, photocatalytic silicon nanosheets derived from rice husks [49] offer an environmentally friendly and economically viable approach to creating functional nanomaterials. Silicon quantum dots (Si-QDs) possess unique optical, electrical, magnetic, and thermal properties, making them promising for use in photocatalytic processes due to their non-toxicity, environmental safety, and widespread occurrence in nature [50]. Additionally, the heterointegration of Pt/Si/Ag nanowires [51] enables the creation of effective hybrid structures that enhance charge transfer and improve PC stability. For effective photocatalytic decomposition of pollutants, as well as for water splitting processes, an optimized interaction between photoactive materials and catalytic components is required. However, a single silicon photoanode or photocathode is unable to provide complete water splitting without an external voltage. At the same time, a system with two photoelectrodes can perform photoelectrochemical (PEC) water splitting without requiring an additional bias, making it more promising for practical applications [52]. Despite numerous advantages, unresolved issues especially with uniformity and state-of-art synthesis persist regarding the improvement of stability, charge transfer processes, and the development of effective co-catalysts.

## 3. Protective Coatings, Nanostructures, and Specialized Applications of Silicon in Photocatalysis

Si nanostructures, such as nanowires (SiNWs) and nanotubes (SiNTs), significantly influence photocatalytic activity by increasing surface area and optimizing light absorption. SiNWs with a length of 9 μm and a diameter of 20–100 nm reduce light reflection to 1% in the 400–800 nm range [53]. Porous silicon with a surface area of 2–20 m^2^/g and 4% reflectivity forms hydroxyl radicals (2.38 eV), effectively decomposing organic pollutants into CO_2_ and H_2_O [54]. The comparison shows that SiNTs achieve 90% degradation of rhodamine 6G in 5 h, while SiNWs achieve less than 70% degradation in the same time, highlighting the role of morphology [55]. Decorating SiNWs with reduced graphene oxide (rGO) increases the photocurrent by a factor of 4 compared to pure SiNWs and by a factor of 600 compared to flat Si/rGO, minimizing recombination and reducing reflection to 4% in the 200–1000 nm range [56,57].

Specific applications of Si include silicon carbide (SiC) and photonic crystals (SiPCs). Since Si is susceptible to corrosion, 3C-SiC, being a chemically stable material, protects the silicon substrate from the destructive effects of aggressive alkaline electrolytes [58]. SiC obtained from solar cell waste is combined with AgCl, addressing the issues of photocorrosion and a wide forbidden zone, thereby making it suitable for photocatalysis [59,60]. The mechanism of water splitting on SiC involves both photocatalytic processes and direct chemical reactions, although the formation of oxide layers slows down the reaction [61]. Two-dimensional silicon SiPCs with cylindrical pillars enhance light absorption (~450 nm) due to the photon bandgap. At the same time, the integration of PtNPs enhances the photocurrent and selectivity for methane (~25% Faradaic efficiency) during the photoelectrochemical reduction of CO_2_ [62]. Figure 2 shows two-dimensional photonic crystals with circular dielectric pillars arranged in a square array (Figure 2A). While keeping the ratio of the pattern radius to the period (r/P ~ 0.3) and the height of the dielectric column unchanged, their periods (P) varied from 1 to 2 μm. The SEM images (Figure 2B) are consistent with the designed patterns. The structural color (Figure 2C) displayed under illumination and the peak part of the absorption spectrum (Figure 2D) showed its modulation effect on a specific wavelength band (~450 nm), further confirming the formation of a photonic crystal structure [62].

Silicon chalcogenides (SiX, X = S, Se, Te) with a forbidden zone of 2.43~3.00 eV and high charge mobility allow properties to be tuned through mechanical stress, which is promising for scalable systems [63]. Silicon quantum dots enhanced with Fe_3_O_4_ enable the complete decomposition of melamine in 20 min under visible light and retain their activity for 15 cycles, thanks to their magnetic properties, which facilitate regeneration [64].

Radial p-n junctions in SiNWs minimize recombination due to perpendicular charge transport. The peril of this kind of material is the requirement of protective coatings, such as silicon nitride, and complex synthesis methods, which increase the cost [65]. The Si-α-Fe_2_O_3_/In_2_S_3–_3 composite achieves 82% degradation in 60 min; however, oxygen vacancies reduce its durability during intensive use [66]. These systems underscore the importance of striking a balance between efficiency, stability, and scalability in silicon-based photocatalytic applications.

Silicon parts in PC systems often require protective coatings to prevent photocorrosion, especially in aggressive aqueous environments. Metals with high work function, such as Ag, AuPd, and Pd, form a Schottky barrier at the Si interface, creating a built-in electric field that improves charge carrier separation [67]. For example, mesoporous SiNWs decorated with Ag and AuPd exhibit enhanced photodegradation of methylene blue under visible light, attributed to the Si-H bonds and efficient electron transfer. However, the loss of these bonds upon Ag decoration reduces the efficiency [67] (Figure 3). Pd/TiO_2_/Si nanopillars synthesized by atomic layer deposition utilize hot electrons for hydrogen and oxygen evolution reactions, providing a synergistic effect across the entire spectrum of solar radiation [34]. Cobalt phosphate (Co–Pi) and oxides, such as indium tin oxide, reduce the threshold potential of silicon photoanodes, thereby improving water oxidation, whereas polymers, like polyvinylidene fluoride, are less resistant to photogenerated charges [68].

## 4. Silicon-Based Heterostructures for Photocatalysis

In recent decades, research on photocatalytic processes has focused on the design of highly efficient and durable photocatalytic materials [69]. The photocatalytic deg-radation of organic dyes depends on their molecular structure and their ability to in-teract with reactive oxygen species generated by light acting on the photocatalyst [70]. Methylene blue (MB) with chromophoric aniline group and labile structure is easily degraded at minimal energy, undergoing demethylation and decomposition into small molecules, which leads to significant pollution of water bodies, as about 10% of the dye is lost in wastewater during dyeing process MB is toxic, negatively affecting flora and fauna through consumption or absorption, and potentially jeopardizing human health when contaminated water is used. During photocatalytic degradation, MB is effective-ly degraded through oxidation by free radicals (O_2_^−^, H_2_O_2_, OH^−^) produced upon activa-tion of a sensitizer (e.g., chlorophyll) [71]. Rhodamine B (RhB) degrades more slowly, forming unstable colorless intermediates that result in discoloration of the solution. Methyl orange (MO), which features a stable azo- and quinoline structure, requires considerable energy for demethylation and reduction of the azo group, thereby reduc-ing its degradation efficiency [72]. N. Sun and co-authors identified six key strategies to enhance their activity, including morphology and crystal structure modification, doping, plasmonic metal nanoparticle deposition, heterostructure design, dye sensiti-zation, and organic modification [73]. Another challenge in the area of azoic dyes is the stability of the catalytic substrate. For this purpose, Si-based heterostructures are a promising candidate [74].

Silicon in multicomponent heterostructures significantly increases PC activity due to synergistic effects caused by a narrow-forbidden zone (1.1 eV) and a more negative conduction band position. The use of Si as a core provides mechanical stability to the composite, preventing particle aggregation and improving the distribution of active centers. Silicon, thanks to its unique optical and electrical properties, acts as an effec-tive carrier for electron transfer, enhancing photocatalytic activity [75] (see Table 1). For example, the Si/MgTiO_3_ heterostructure exhibits improved activity compared to individual components, owing to effective charge separation and an extended light ab-sorption spectrum [76]. Similarly, Si/TiO_2_ composites containing nanosilicon increase the efficiency of methylene blue degradation under UV radiation due to improved elec-tron transfer and increased surface area [77]. By the way, a larger number of examples reported in Table 1 demonstrate efficiency under real or simulated sunlight. This makes silicon-oxide interfaces promising candidates for real-world applications de-spite the performance being inferior to more sophisticated materials, which will be dis-cussed below. 

Recycled silicon solar cells could be a source of silicon for these heterostructures. The application of recycled silicon solar cells is a charge transfer bridge in Z-schemes [78]. Recycling solar panel waste to create TiO_2_/Si photocatalytic carriers reduces en-vironmental and economic costs, although high-temperature processing limits the scalability of such systems [79]. Chemical methods for the production of silicon nanostructures from end-of-life silicon solar cells are also relatively costly [80]. Thus, other applications, such as construction, are more economically and environmentally sustainable [81]. 

**Table 1 nanomaterials-15-01514-t001:** Photocatalytic properties of Si-based nanostructures.

Photocatalysts	Object of Decomposition	Photocatalytic Parameters	Light Source	Source
Si/TiO_2_ nanotubes (500 nm length, 80 nm diameter, 16 nm wall thickness)	Rhodamine B	~1.78 times higher kinetic constants compared to TiO_2_ nanotubes	UV	[82]
Si substrates with Ag nanostructures	Methyl Orange	Rate constant 33.5 × 10^−3^ min^−1^	sunlight	[83]
Si/SiC@C@TiO_2_	Methylene Blue	Rate constants 4.5 (UV) and 7.9 (visible) times higher than TiO_2_	UV, simulated solar	[84]
Sulfonated mesoporous silica/ZnO	Methylene Blue	97.419% efficiency	simulated solar	[85]
SBA-16/TiO_2_	Paraquat herbicide	Si/Ti ratios 5.6, 1.4, 0.7, complete decolorization under UV, 70% PQ reduction in 24 h by 1.4 ratio	UV	[86]
Fe_3_O_4_/PDA/Si-Ca-Mg (FPS)	Methylene Blue	Adsorption capacity 100.23 mg/g, recyclable adsorbent	sunlight	[87]
Cu-Si Nanoparticles	Methylene Green	Photocatalyst doses: 10 mg (46.4%), 20 mg (81.7%), 30 mg (95.7%) degradation efficiency	sunlight	[88]
TiO_2_/RH-SBA-15	Methyl Orange	30% TiO_2_ ratio, 50 ppm initial dye concentration, 200 mg catalyst, 63% higher efficiency than bare TiO_2_	simulated solar	[89]
V(0.005)-NSiT, V(0.02)-NSiT	DMSO	40% (V(0.005)) and 29% (V(0.02)) DMSO decomposition in 10 h under visible light	simulated solar	[90]

The mechanism of the photocatalytic reaction based on Si NWs/ZnO photocatalysts is illustrated in Figure 4. Under the action of light, Si NWs absorb photons and generate electron-hole pairs:(1)SiNWs + hν→SiNWs (e^−^, h^+^)

A similar process occurs in ZnO:(2)ZnO + hν→ZnO (e^−^, h^+^)

Further interaction between Si NWs and ZnO leads to charge redistribution, in which electrons remain in Si NWs and holes are transferred to ZnO:(3)SiNWs (e^−^, h^+^) + ZnO (e^−^, h^+^)→SiNWs (e^−^) + ZnO (h^+^)

The freed holes participate in oxidative reactions. Interaction with water molecules leads to the formation of hydroxyl radicals, which have high reactivity:(4)H_2_O + h^+^→⋅OH + H^+^

The oxidation of organic pollutants, such as rhodamine B (RhB), leads to its decomposition with the formation of carbon dioxide and water:(5)RhB + h^+^→CO_2_ + H_2_O

At the cathode, oxygen is reduced, accepting electrons and protons to form water molecules:(6)O_2_ + 4e^−^ + 4H^+^→2H_2_O

This reaction scheme demonstrates how SiNWs/ZnO promote the photocatalytic decomposition of organic compounds and participate in oxidation and reduction processes. Charge transfer between materials helps reduce the likelihood of electron-hole pair recombination, and the resulting active oxygen species participate in the oxidation of pollutants, making this system an effective means of water purification [2].

Si/ZnO heterostructures are also promising, especially in the visible range. The addition of Si nanocrystals to ZnO nanostructures enhances photocatalytic activity by narrowing the ZnO band gap (from 3.22 to 3.07 eV) and increasing light absorption, resulting in a 15% increase in the degradation rate of methylene blue under white light [91,92]. When p-Si and n-ZnO come into contact, a p-n junction is formed, which equalizes the Fermi levels, leading to band bending and the creation of an internal electric field at the interface. This field effectively separates the photogenerated electrons and holes, preventing their recombination [93]. The ZnO/Cu_2_O/Si heterostructure, synthesized by deposition and thermal annealing, provides stepwise separation of energy bands. This led to a methylene blue degradation rate 15.3 times higher than that of pure silicon, and 5.7 and 3.4 times higher than that of the binary systems ZnO/Si and Cu_2_O/Si, respectively, with second-order kinetics (R^2^ = 0.98) [94]. SiNWs/TiO_2_ composites outperform pure TiO_2_ in dye degradation, such as RB5, due to improved photon absorption at longer wavelengths, which is associated with the effective utilization of wide-spectrum radiation. [95]. TiO_2_/SiNWs heterostructures demonstrate improved photocatalytic properties in the decomposition of organic dyes. To synthesize such systems, silicon nanowires were obtained by galvanic displacement and then coated with TiO_2_ nanoparticles by spraying [96]. Optimizing the interaction between silicon and TiO_2_ requires precise tuning of coating thickness and composition to balance electron and hole barriers [97]. Silicon increases the surface area of TiO_2_ and prevents the phase transition from anatase to rutile, thereby improving the stability of the photocatalyst [96,98]. Nanoparticles of 15% Si-TiO_2_ showed 98% degradation of methyl orange in 40 min under UV light, outperforming commercial TiO_2_ Degussa P25 (84% in the same time). Si content above 15% leads to a decrease in photocurrent (e.g., 25% Si-TiO_2_ has a photocurrent of only 12.9 μA), as excess Si can act as a recombination center for electrons and holes [99]. Table 2 contains a summary of the recent advantages of photocatalytic NWRs based on Si in organic pollutants removal processes. As one can see, the best performance of these materials corresponds with UV irradiation. However, the results obtained for materials demonstrates potential of Si-based NWRs for photdegradation of organic compounds.

Besides standard compounds used for testing the feasibility of pollutants’ degradation, several silicon-based systems were used for real-world contaminants. Sulfur- and nitrogen-doped silicon was used for efficient degradation of tetracycline [100]. Successful employment of silica-TiO_2_ composites for photo-degradation of water-soluble pharmaceuticals [101] and pesticides [102,103] demonstrates the potential of the oxidized surface of Si-nanosystems for this purpose.

**Table 2 nanomaterials-15-01514-t002:** Summary of photocatalytic Si-based NWRs.

Si Nanowires Description	Dye Degraded	Photocatalytic Parameters	Light Source	Source
Si nanowires combined with ZnFe_2_O_4_/Ag.	Methyl Orange	Degradation rate: 19% in 90 min.	UV	[104]
Si nanowires coated with TiO_2_ and MoS_2_ nanosheets.	Rhodamine B	Degradation efficiency ~ 90% within 180 min.	UV	[105]
Si nanowires mixed with TiO_2_ microparticles.	Remoazol Black 5	Degradation efficiency 50.9% after 150 min of continuous 580 nm.	simulated solar	[95]
Si nanowires modified with Au, Pt, Pd nanoparticles	Methylene Blue	Pd-modified SiNWs: degradation rate of 97% after UV irradiation 200 min.	UV	[106]
Si nanowires (23–30 µm length) decorated with CoO, Cu, Ag nanoparticles.	Methyl Orange	Degradation: Si-NWs-Cu-NPs (88.9%), Si-NWs (85.3%), Si-NWs-CoONPs (49.3%).	simulated solar	[107]
Si nanowires decorated with Ni-doped ZnO	Methylene Blue	97% degradation efficiency with 5% Ni-ZnO/SiNWs	UV	[108]
Si nanowires (2.5–13.5 µm) coated with TiO_2_ nanoparticles	Methylene Blue	96% degradation efficiency with 3.5 µm nanowire length; stable after 190 days.	UV	[96]
Si nanowires (80–100 nm diameter)	Rose Bengal	~96% degradation in ~90 min under light illumination.	sunlight	[109]
Si nanowires decorated with Bi nanoparticles	Methylene Blue	44% degradation under UV and 89% under solar irradiation in 120 min.	UV, sunlight	[110]
Si nanowire arrays prepared by metal-assisted chemical etching with varying H_2_O_2_ concentrations.	Rhodamine B	35% degradation after 5 h of irradiation for 20% H_2_O_2_	simulated solar	[111]
Si nanowires (1–42 µm length) modified with graphene oxide.	Methylene Blue	92% degradation at 10 min etching, 4:1:8 etchant ratio; 96% with H_2_O_2_; bare Si (16%), GO/bare Si (31%).	UV	[112]
Si nanowires coated with CeO_2_ nanoparticles	Rhodamine B	Quasi-total discoloration in 75 min; 67% for bare Si-NWs; involves e^−^, •OH, O_2_•^−^, h^+^ species.	sunlight	[113]

The use of carbon nanotubes (CNTs) in the Si/CNTs composite prevents charge carrier recombination, improving photocatalytic efficiency. However, after several cycles, a decrease in activity is observed due to the formation of Si-OH, which limits the process’s efficiency [114]. Complex heterostructures, such as Co_3_O_4_/Si nanoarrays and epitaxial NiO(111)/c-YSZ(001)/Si(001), improve catalytic performance through effective charge separation and broadening of the absorption spectrum [115,116]. For example, SiNWs decorated with Co_3_O_4_ show increased activity due to reduced electron recombination, although silicon is inferior to TiO_2_ in terms of resistance to photocorrosion [117]. The MoS_2_/TiO_2_/SiNWs composite increases the photodegradation rate of rhodamine B by 60 times compared to TiO_2_/SiNWs, but hydrothermal treatment (180 °C, 24 h) can cause structural changes that limit durability [105]. The Si/Cu_2_O/CNT heterostructure, which features carbon nanotubes connecting silicon pillars, achieves an 86% degradation of methylene blue in 2 h under visible light, primarily due to the internal electric field and increased surface area [39]. Figure 5 show a schematic representation of the formation process of the Si/Cu_2_O/CNT heterostructure.

Z-scheme photocatalysts, such as Si-SnO_2_-TiO_x_ (1 < x < 2), use transparent SnO_2_ as an electron mediator, increasing the photocurrent by an order of magnitude and achieving 75% phenol decomposition without external bias, as well as 70% removal of total organic carbon [118]. The SiNWs/ZnO heterostructure in photocatalytic fuel cells effectively decomposes rhodamine B (with 93% efficiency) and generates electricity (current density of 0.183 A/m^2^, power of 0.87 W/m^2^), maintaining stability over 20 cycles [2]. These systems demonstrate the potential of silicon in developing sustainable and effective photocatalysts, although issues of photocorrosion and interface complexity necessitate further optimization [104,119].

## 5. Machine Learning in the Optimization of Photocatalysts

Traditional methods for calculating redox potentials using Density Functional Theory (DFT) are computationally complex, and their accuracy depends on the choice of functional, particularly for excited states [120,121]. Machine learning (ML) enables calculations to be accelerated, thereby reducing computational costs and enhancing accuracy [122]. A combined approach that integrates DFT and ML provides accuracy comparable to experimental data with fewer resources [122]. For example, Gaussian approximation potentials (GAP) demonstrate high accuracy in modeling the thermal conductivity of crystalline and amorphous silicon, surpassing empirical potentials and approaching DFT with a significant reduction in computational costs [123,124]. ML is also effective in analyzing images of electroluminescence and photoluminescence. Convolutional neural networks (CNNs) and the YOLO (You Only Look Once) algorithm accurately detect defects, including cracks, degradation, and contamination. Infrared thermography and ultraviolet fluorescence are used to evaluate the performance of silicon photovoltaic modules [125]. Thus, ML is an important tool to help in the interpretation of raw data.

The integration of ML with domain knowledge of photocatalysis opens up prospects for the creation of rapid screening platforms for photocatalysts. Usually, the aim of these studies is the design of novel compounds with desirable values of the bandgap and position of the valence and conductive bands edges [126,127]. This approach addresses the issue of data scarcity, enhances the reliability and interpretability of models, and contributes to the development of effective materials for solar technologies [128]. For example, ML has been successfully applied to the design of ABO_3_ perovskite-based photocatalysts. The support vector regression (SVR) algorithm, integrated with a radial basis function and a web service, accurately predicts the specific surface area (SSA) by identifying key factors that influence synthesis [129]. ML models, such as gradient boosting regression (GBR), effectively predict the degradation of methylene blue (MB) based on Al and Ag_3_PO_4_ content. GBR identified optimal compositions with a minimum number of experiments, demonstrating that the Ag_3_PO_4_ content has a more substantial influence on effectiveness than the Al content [130].

ML methodology for limited data proposed by Soltani et al. [131], includes four stages: collecting a dataset of 31 articles (597 data points) on stanine and hydroxystanine photocatalysts, creating descriptors (32 photocatalyst parameters and 1024-bit molecular fingerprints of pollutants), selection of 16 key features, and training of Random Forest (RF) and KNN models. RF showed high accuracy (R^2^ = 0.943, MAE = 0.0918). SHAP analysis highlighted the importance of dosage, pH, and irradiation time. The model was successfully tested on CdSnO_3_ and a new contaminant (crystalline violet) [128]. Convolutional neural networks for crystal graphs (CGCNN), combined with molecular fingerprints and artificial neural networks, effectively predict the rate constants of photocatalytic degradation of oxide photocatalysts and organic pollutants (R^2^ = 0.746). The model using SHAP analysis is scalable and applicable for optimizing wastewater treatment [132]. Random Forest and ANN algorithms optimize the doping and morphology of photocatalysts to improve efficiency [133]. Zhou et al. [134] utilized PyMatGen to generate a library of 800 oxides, accurately predicting band gaps and identifying CsYO_2_ as a promising material for water splitting [134]. ML also optimizes the production of atomic qubits based on phosphorus donors in silicon [135] and creates regular nanopores in silicon using SVM, achieving an accuracy of up to 98% and identifying key parameters such as H_2_O_2_ volume [136].

For amorphous silicon (a-Si), widely used in photovoltaic devices, ML potentials in the teacher-student approach provide DFT accuracy (~10 meV/atom) with a speedup of 1000 times. Defect analysis (*n* = 3, *n* = 5) using SOAP, t-SNE, and clustering reveals their energetics and structure, confirming agreement with experimental data [137]. The Atomic Cluster Expansion (ACE) model accurately describes the Si-O system (Si, SiO_2_, SiO), reproducing segregation and crystallization, which is important for semiconductors and solar cells [138]. Highly efficient search for direct-bandgap silicon allotropes using ML potentials and DFT revealed 47 structures, 22 of which are promising for photovoltaic applications. The Si12-P1 structure, with a direct band gap of 1.69 eV, achieves a maximum spectroscopic light extraction efficiency (SLME) of 32.28%, surpassing that of diamond silicon [139].

The application of machine learning in the optimization of photocatalysts significantly accelerates material design, reducing computational costs and improving prediction accuracy compared to traditional DFT methods. Algorithms such as Random Forest, SVR, CNN, and GBR demonstrate high effectiveness in predicting key properties of photocatalysts, including redox potentials, specific surface area, and degradation rate constants. In summary, these methods successfully solved the limitation of DFT-based methods in the number of atoms, which is essential for the simulations of realistic nanosystems and surfaces with complex morphologies.

The integration of ML with domain knowledge and experimental data enables the creation of scalable platforms for material screening, facilitating the development of innovative solutions for solar energy and wastewater treatment. However, the limited availability of datasets, which mainly comprise successful experiments, necessitates further model improvement to enhance their generalizability and reliability in real-world conditions. The key difficulty with building datasets suitable for ML requires the acceptance of a standardized description of experimental conditions and results. In current situations, the key experimental data for this type of dataset extracted manually, which limits size of these datasets.

## 6. Silicon Doping to Enhance Photocatalytic Activity

Silicon doping significantly enhances the photocatalytic properties of semiconductors, such as TiO_2_, ZnO, and hematite, by modifying their electronic structure and optical characteristics. The introduction of Si into the material structure creates intermediate states in the forbidden zone, reducing its width and increasing the density of electronic states near the Fermi level, which promotes light absorption in the visible range [140]. For example, Si-doped TiO_2_ demonstrates a reduction in the band gap width from 3.13 to 2.96 eV, which increases the efficiency of electron and hole separation, increases the surface area, and provides 96% degradation of methyl orange and crystal violet under sunlight, outperforming F-doped and undoped TiO_2_ [141]. Studies confirm that Si doping enhances photocatalytic activity by reducing charge carrier recombination centers and increasing the density of hydroxyl groups on the surface [142,143,144].

However, the concentration of silicon must be carefully controlled. Insufficient doping has a minimal effect on activity, while excessive Si content shields the active centers, reducing the efficiency of photocatalysis, for example, in water purification [145]. To enhance the effect, co-doping is used. Joint doping of TiO_2_ with silicon and nickel creates a synergistic effect: isolated Ni 3*d* states form an acceptor level below the conduction band edge, and the narrowing of the forbidden zone broadens the absorption spectrum [146,147]. N,Si-codoped TiO_2_, according to Z. Ai et al., decomposes salicylic acid under visible light (λ > 420 nm) with a rate constant of 11 × 10^−2^·h^−1^, which is 5 times higher than that of pure TiO_2_, due to the formation of Ti–O–Si bonds that improve charge transfer [148]. Theoretical calculations indicate that Si mixes s-orbitals with Ti 3*d*-orbitals, altering the position of the conduction band edge of TiO_2_, which in turn enhances the photocatalytic activity [149].

Doping hematite (α-Fe_2_O_3_) with silicon increases the proportion of active (110) faces and creates oxygen vacancies, reducing the width of the forbidden zone and improving photocatalytic properties [150]. Similarly, Si-doped polymeric carbon nitride (CNSi) exhibits a hydrogen evolution rate of 2.24 mmol·g^−1^·h^−1^, nearly three times higher than that of the undoped material, with a quantum yield of 7% at 420 nm [151]. Si doping also affects the magnetic properties of materials due to the spin polarization of silicon 3*p* orbitals and interactions with neighboring atoms, which may be helpful for multifunctional photocatalysts [152]. For example, adding Gd^3+^ and Si^4+^ to NaYF_4_ creates sub-energy levels, extending light absorption into the UV-visible region [153].

Despite its advantages, Si doping has limitations. The high light absorption coefficient of silicon can reduce the number of photons available to the active photocatalytic layer, which decreases the generation of active oxygen species [154]. In addition, SiO_2_, which is often used as an auxiliary material, does not have photocatalytic activity on its own. Still, its mesoporous structure effectively immobilizes TiO_2_ nanoparticles, controlling their size and improving light scattering in the system [155,156]. A comparative analysis reveals that Si doping is most effective in systems that require operation in the visible spectrum. Still, additional modifications are necessary for the UV range, such as plasmonic nanoparticles or visible-sensitive components [157]. The method for creating Ti-Si binary oxides proposed by Masato Takeuchi and co-authors demonstrates significant potential for improving photocatalytic properties through the interaction of Ti oxides with SiO_2_. At the same time, high stability and selectivity make Ti-Si binary oxides a promising material for controlling NO_x_ emissions under UV illumination [157]. Thus, silicon doping opens up broad possibilities for optimizing photocatalysts, but requires balancing the concentration and considering the optical properties of the material. Table 3 presents a summary of the recent advantages of Si-doping in photocatalytic lysis for the decomposition of organic pollutants.

## 7. Conclusions and Outlooks

Silicon nanostructures demonstrate significant potential in photocatalytic applications due to their low cost, availability, tunable bandgap, and ability to form effective heterostructures. Synthesis methods, such as metal-assisted chemical etching and hydrothermal approaches, enable the creation of nanostructures with high surface areas and improved optical properties. Heterostructures, such as Si/ZnO and Si/TiO_2_, significantly increase photocatalytic activity through synergistic charge separation and broadening of the light absorption spectrum. Doping semiconductors, such as TiO_2_, with silicon optimizes their electronic structure, thereby increasing their efficiency in the visible range. Photocorrosion and degradation of activity due to surface oxidation remain key limitations requiring the development of protective coatings and co-catalysts. On the other hand, the silica-like surface phase can be the source of additional optical transitions.

Overall, Si-based photocatalysts have high potential for addressing environmental challenges, but their practical implementation requires further optimization of stability and scalability. Future research on Si-based photocatalysts should focus on solving the problem of photocorrosion by developing stable protective coatings, such as silicon nitride or metal oxides, that retain high photocatalytic activity. Optimizing heterostructures, including Z-scheme systems, will enhance charge separation efficiency and broaden the light absorption spectrum. The integration of plasmonic nanoparticles and co-catalysts, such as Pt or Pd, can further enhance catalytic properties. The current aim in this area is to transition from systems that are efficient under UV radiation to systems that can utilize sunlight with a similar level of efficiency. Another important challenge is the turning of the objects of studies from molecules like methylene blue to the real world contaminants (pharmaceuticals, personal care products etc.).

Particular attention should be paid to the development of environmentally friendly and cost-effective synthesis methods, such as utilizing waste from the solar industry. Research into silicon quantum dots and two-dimensional nanostructures such as silicene will open up new possibilities for photocatalysis and photoelectrochemistry. Scaling up laboratory developments to industrial levels will require an interdisciplinary approach combining nanoengineering and computational methods. Utilization of out-of-life solar cells and perhaps computer chips for making Photocatalysts is another prospective direction.

The application of machine learning, including Random Forest algorithms and convolutional neural networks, accelerates the design of photocatalysts by helping with the recognition and interpretation of raw data. On the other hand, the multiplicity of Si-based systems with different macroscopic and microscopic morphologies, a long list of materials for the formation of heterojunctions and their morphologies, makes the ML approach essential for guided design of the efficient Si-based photocatalysts produced by environment-friendly techniques. The application of ML in this complex area requires the development of standardized datasets of experimental conditions and results of the measurements.

## Figures and Tables

**Figure 1 nanomaterials-15-01514-f001:**
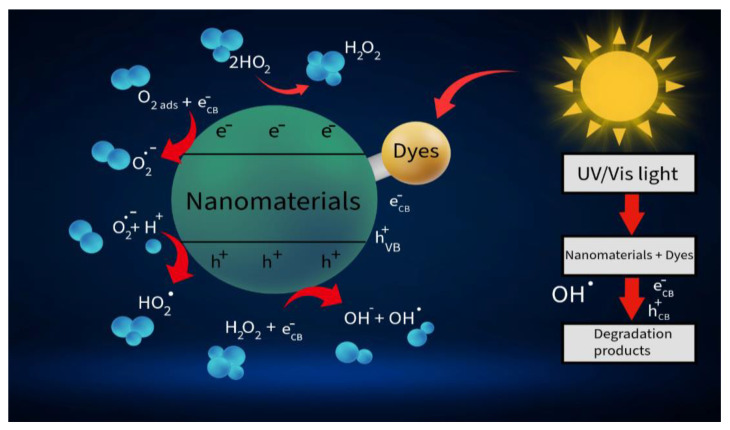
Schematic illustration of the dye degradation mechanism (Adopted from [11] with permission from Elsevier).

**Figure 2 nanomaterials-15-01514-f002:**
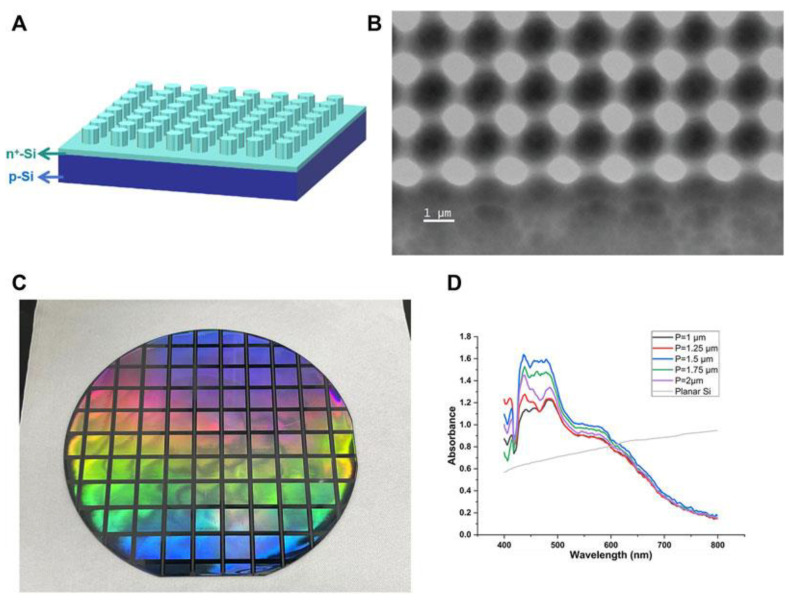
(**A**) Schematic of silicon photonic crystals (SiPCs). (**B**) SEM image of the SiPC. (**C**) Digital photo. (**D**) Absorption spectra. Reproduced from [62] distributed under the terms of the Creative Commons Attribution License (CC BY).

**Figure 3 nanomaterials-15-01514-f003:**
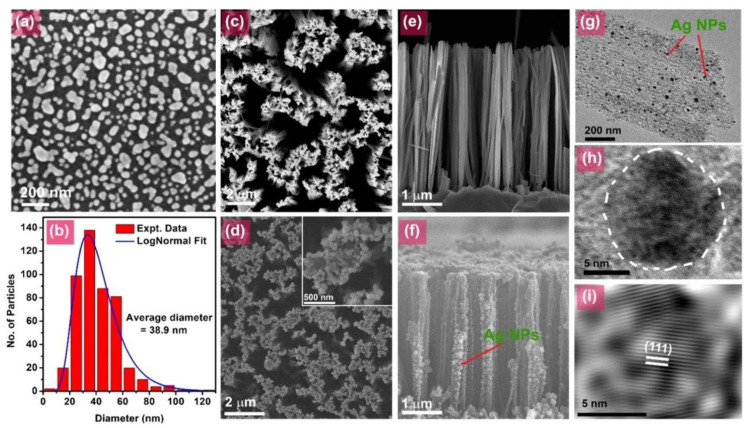
(**a**) FESEM top view image of SiAg1. (**b**) Size distribution of Ag NPs with a lognormal approximation. FESEM images: (**c**) NW and (**d**) NWAg1 with enlarged inset. (**e**,**f**) Cross sections of NWs, with Ag NPs marked by arrows. (**g**) TEM image of Si NW with Ag NPs. (**h**) HRTEM image of Ag NP crystal lattice on the surface of Si NW. (**i**) Enlarged IFFT image confirms the (111) plane of Ag (Reprinted from [67] with permission from Elsevier).

**Figure 4 nanomaterials-15-01514-f004:**
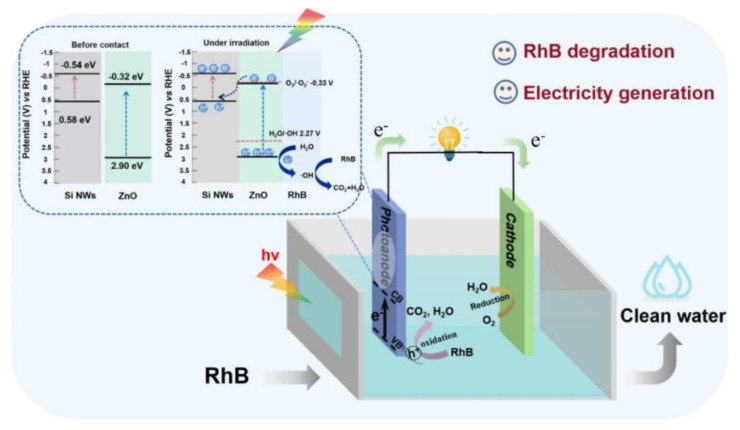
Working mechanism of photocatalytic fuel cell based on the Si NWs/ZnO heterojunction photoanode (Reprinted from [2] with permission from Elsevier).

**Figure 5 nanomaterials-15-01514-f005:**
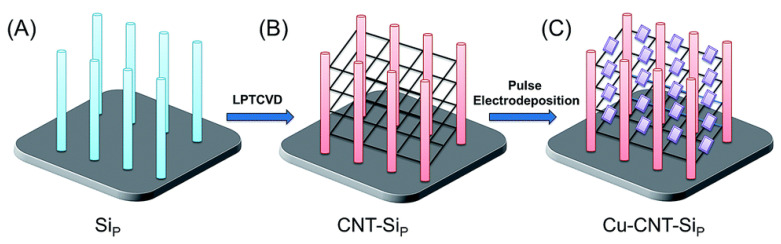
Graphic illustration of each step for the preparation of suspended cuprous oxide (Cu_2_O) architecture. Reproduced from [39] distributed under the terms of the Creative Commons Attribution License (CC BY).

**Table 3 nanomaterials-15-01514-t003:** Si-doped structures with quantitative characteristics of photocatalysis.

Structure	Object of Decomposition	Photocatalyst Parameters	Light Source	Source
Si-doped TiO_2_ nanotubes	Methyl Orange	5% Si-TiO_2_ NTs showed much higher photocatalytic activity	UV	[158]
Si-doped TiO_2_ nanotubes	Methylene Blue	10% Si-doped TiO_2_ nanotubes tripled MB degradation efficiency compared to undoped TiO_2_	UV	[159]
Si-doped TiO_2_ nanotubes	Phenol	10% Si-doped TiO_2_ nanotubes showed ~9 times higher phenol degradation under visible light compared to undoped TiO_2_.	simulated solar	[160]
TiO_2_/SiO_2_ (Si-doped TiO_2_ in PDMS microreactor)	Methylene Blue	Degradation efficiency: 93.59% after 90 min	UV	[161]
TiO_2_-SiO_2_ (Si-doped TiO_2_ hybrid)	Rhodamine B	Degradation efficiency: 95.81% for 60 mg/L RhB over 3 cycles-High surface area from biochar/zeolite support	UV	[162]
TiO_2_-SiO_2_ (Si-doped TiO_2_ monolith)	Phenol	Degradation efficiency: 92% after 240 min with PMS	UV	[163]
CdS-BiVO_4_ (Si-doped artificial leaf)	Rhodamine B	Degradation efficiency: 92% after 2 h visible light irradiation (2.1 times higher than no-template BiVO_4_)	sunlight	[164]

## Data Availability

No new data were created or analyzed in this study.

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
