# Peer review of "Silicon Photocatalytic Water-Treatment: Synthesis, Modifications, and Machine Learning Insights"

_nanomaterials, 2025, doi:10.3390/nano15191514_

Round 1
Reviewer 1 Report
Comments and Suggestions for Authors
File attached.

Author Response
Reviewer 1
Review Report:
In the present work titled "Silicon Photocatalytic Water-Treatment: Synthesis, Modifications, and
Machine Learning Insights" authors presents a review of silicon (Si)-based photocatalysts,
focusing on their application in water treatment, hydrogen generation, and CO2 reduction. It
discusses various synthesis methods, including chemical deposition, metal-assisted etching,
hydrothermal methods, and atomic layer deposition. The paper highlights the use of
heterostructures, plasmonic effects, and co-catalysts in enhancing photocatalytic activity.
Additionally, the application of machine learning (ML) for optimizing photocatalyst performance
is explored. This article can be published after careful revisions.
Answer:
Thank you for the high evaluation of our work and valuable comments. Please find below, line-by-line, answers to your questions.
Reviewer:
1. The manuscript primarily summarizes well-known methods and materials, offering
minimal new insights or significant contributions to the field.
Answer:
The review papers aim to summarize and compare the results. The multiple references to the works published in the last five years demonstrate that the area is still developing. To enhance the level of novelty, we increased discussion of the recent works in the revised manuscript.
Reviewer:
2. The review lacks in-depth comparison and critical evaluation of methods. There is little
discussion of challenges, limitations, or gaps in the field.
Answer:
We improved the methods-related section in the revised manuscript. At the end of each paragraph devoted to the description of the methods, the limitations were discussed. Some “gaps in the fields”, such as overuse of UV radiation and underreporting of the results for actual sunlight, were pointed out. The addition of the light-source-related columns in the tables pointed out the latest issue.
Reviewer:
3. The manuscript mentions potential applications but fails to explore real-world scalability,
economic feasibility, or environmental impact in detail.
Answer:
We enhanced the discussion of these points in the revised manuscript. The advantages and perils of scaling up each technique described in the second section were discussed in the concluding sentences of each paragraph devoted to different methods.
Reviewer:
4. Machine learning is mentioned but not explored effectively. Specific case studies or
tangible examples of ML applications in photocatalysis are missing.
Answer:
We are surprised by these comments because in the ML, only two references are older than five years, and eight references are dated by this or previous years. By the way, some recent works on applying ML in photocatalysis have been discussed.
Reviewer:
5. Proposed solutions for issues like photocorrosion and surface oxidation are standard and
do not present innovative advancements.
Answer:
We extended the discussion of potential impact of surface oxidation in photocatalysis.
“By the way, a larger number of examples reported in Table 1 demonstrate efficiency under real or simulated sunlight. This makes silicon-oxide interfaces promising candidates for real-world applications despite the performance being inferior to more sophisticated materials, which will be discussed below.
…
Besides standard compounds used for testing the feasibility of pollutants’ degradation, several silicon-based systems were used for real-world contaminants. Sulfur- and nitrogen-doped silicon was used for efficient degradation of tetracycline (Wang et al., 2022). Successful employment of silica-TiO2 composites for photo-degradation of water-soluble pharmaceuticals (Gusmao et al., 2022) and pesticides (Kalidhasan and Lee, 2022; AbuKhadra et al., 2020) demonstrates the potential of the oxidized surface of Si-nanosystems for this purpose.”
Reviewer:
6. Some references, especially related to machine learning in photocatalysis, are outdated and
should be replaced with recent studies.
Answer:
As we pointed out above, in the ML section, only two references from the dozen are older than five years. In the other sections, we also cited many works published in current or recent years. We partially agree with the reviewer regarding the outdated references, as comparing recent and previous results is essential for tracking progress in the area. However, we added more than new references to the works published in this or prior years.
Reviewer:
7. The writing is dense and sometimes unclear. Simplifying complex sections would improve
readability.
Answer:
Multiple sentences and some paragraphs were rewritten to make reading easier.
Reviewer:
8. Including figures or tables comparing synthesis methods and their effectiveness would
improve the manuscript.
Answer:
The synthesis methods are strongly associated with the final products (doped silicon, heterojunctions, etc.), as noted in the revised section two. Therefore, the different types of materials discussed in the following chapters were synthesized by different methods and no further corrections is necessary.
Reviewer:
9. The environmental implications of Si-based photocatalysts are not sufficiently discussed.
Answer:
Environmental implications, such as recycling out-of-life solar cells and cleaning of water from real-life pollutants (pharmaceuticals, pesticides), were pointed out in the revised manuscript.
Reviewer 2 Report
Comments and Suggestions for Authors
This manuscript presents a review of Si-based photocatalysts for water treatment, with attention to synthesis routes, structural modifications, and the emerging role of machine learning. The manuscript compiles a wide range of studies. However, in its current form, the work suffers from structural, conceptual, and presentation issues that prevent it from reaching the standards expected for Nanomaterials. Substantial revision is necessary.
1.While the introduction correctly describes the generation of reactive oxygen species (ROS), the authors should emphasize that photocatalytic efficiency is not determined only by the bandgap value, but also by whether the conduction and valence band potentials are thermodynamically suitable for Oâ‚‚ reduction and Hâ‚‚O oxidation.
2. The Introduction section devotes excessive space (lines 64–82) to specific details of methylene blue degradation. Such details are more appropriate later; the introduction should provide broader context and the current state-of-the-art rather than diving into case-specific results. I strongly recommend removing this part.
3. The authors should acknowledge that azo dyes can undergo photosensitization, leading to self-degradation while simultaneously sensitizing the catalyst. This is a known phenomenon that should be discussed to avoid over-attributing degradation efficiency to the photocatalyst itself.
4. Section 2 is overloaded with information presented in a somewhat disorganized manner. The synthesis techniques (chemical deposition, MACE, hydrothermal, ALD, reduction methods, etc.) should be grouped into clearer subsections where their influence on morphology and photocatalytic performance is explicitly compared.
5. The authors intermittently present photocatalytic performance values (e.g., quantum yields, degradation rates) without experimental conditions (e.g., light source, irradiance, wavelength range). At minimum, the type of irradiation (UV, visible, simulated solar) should be specified whenever efficiency is reported. Without this context, performance comparisons are meaningless.
6. The recycling of silicon from solar cells is an interesting and sustainable pathway, but it is only briefly touched upon. This topic deserves a more developed subsection to properly situate it within the broader narrative of Si-based photocatalysts for scalable water treatment.
7. The tables summarize several photocatalytic systems but provide limited information on test conditions and pollutants. At present, they are dominated by dye degradation (methylene blue, rhodamine B, methyl orange). To broaden relevance, examples involving pharmaceuticals, pesticides or other emerging contaminants should be included where available.
8. Most referenced studies rely on UV irradiation. Since UV-driven photocatalysis is less practical for real-world application, more emphasis should be placed on visible-light or solar-driven systems. The review should systematically highlight which Si-based systems are active under visible light and compare their potential for scale-up.
9. Section 5. This section is useful but remains descriptive. The authors should expand the discussion by analyzing how ML could be applied not only to predict redox potentials or material stability but also to accelerate high-throughput synthesis and screening of Si-based photocatalysts. The review should also point out the limitations (e.g., scarcity of large, high-quality datasets) and the need for open databases in this field.
10. The conclusion section is currently a summary of previous sections. It should instead articulate: (i) key knowledge gaps (e.g., photocorrosion resistance, scalability, dataset limitations for ML), (ii) challenges for future research (e.g., stability under natural sunlight, testing with real wastewater, integration into hybrid treatment systems), and (iii) opportunities for new research avenues (e.g., machine learning-guided design, valorization of industrial waste streams for Si precursors, coupling photocatalysis with biological treatments).
Minor comments
-
The abstract is generally clear but could better highlight novelty compared to prior reviews on Si-based photocatalysts.
-
Several figures (e.g., Figures 1–3) appear to be re-adapted from other sources. The figure captions should explicitly state permissions/licenses where necessary.
-
Some references are outdated; inclusion of more recent 2023–2025 works on visible-light photocatalysis and wastewater applications would improve the review.
- English grammar and sentence structure are acceptable overall but require polishing in some sections for conciseness and clarity.
Grammar and syntax should be improved through the manuscript.
Author Response
Reviewer 2
Comments and Suggestions for Authors
This manuscript presents a review of Si-based photocatalysts for water treatment, with attention to synthesis routes, structural modifications, and the emerging role of machine learning. The manuscript compiles a wide range of studies. However, in its current form, the work suffers from structural, conceptual, and presentation issues that prevent it from reaching the standards expected for Nanomaterials. Substantial revision is necessary.
Answer:
Thank you for the careful reading of our work and valuable comments. Please find below line-by-line answer to your questions and notes.
Reviewer:
1.While the introduction correctly describes the generation of reactive oxygen species (ROS), the authors should emphasize that photocatalytic efficiency is not determined only by the bandgap value, but also by whether the conduction and valence band potentials are thermodynamically suitable for Oâ‚‚ reduction and Hâ‚‚O oxidation.
Answer:
Thank you, we pointed out this issue in the revisited manuscript:
“The valence and conductive bands potentials are essential for this process (Yang et al., 2025). ”
Reviewer:
2. The Introduction section devotes excessive space (lines 64–82) to specific details of methylene blue degradation. Such details are more appropriate later; the introduction should provide broader context and the current state-of-the-art rather than diving into case-specific results. I strongly recommend removing this part.
Answer:
We agree. This section was moved to the beginning of the fourth section.
Reviewer:
3. The authors should acknowledge that azo dyes can undergo photosensitization, leading to self-degradation while simultaneously sensitizing the catalyst. This is a known phenomenon that should be discussed to avoid over-attributing degradation efficiency to the photocatalyst itself.
Answer:
Thank you, we pointed out this issue in the revised manuscript:
“Another challenge in the area of azoic dyes is the stability of the catalytic substrate. For this purpose, Si-based heterostructures are a promising candidate (Atyaoui and Ezzaouia, 2025).”
Reviewer:
4. Section 2 is overloaded with information presented in a somewhat disorganized manner. The synthesis techniques (chemical deposition, MACE, hydrothermal, ALD, reduction methods, etc.) should be grouped into clearer subsections where their influence on morphology and photocatalytic performance is explicitly compared.
Answer:
The text in the section 2 was divided by paragraphs. In each paragraph was described key approaches, advantages and perils of methods.
Reviewer:
5. The authors intermittently present photocatalytic performance values (e.g., quantum yields, degradation rates) without experimental conditions (e.g., light source, irradiance, wavelength range). At minimum, the type of irradiation (UV, visible, simulated solar) should be specified whenever efficiency is reported. Without this context, performance comparisons are meaningless.
Answer:
Thank you, appropriate columns were added in all tables.
Reviewer:
6. The recycling of silicon from solar cells is an interesting and sustainable pathway, but it is only briefly touched upon. This topic deserves a more developed subsection to properly situate it within the broader narrative of Si-based photocatalysts for scalable water treatment.
Answer:
We extend discussion of the application of recycled silicon as much as as possible:
“Recycled silicon solar cells could be the source of silicon for these heterostructures. The application of recycled silicon solar cells is a charge transfer bridge in Z-schemes (Yuan et al., 2024). Recycling solar panel waste to create TiO2/Si photocatalytic carriers reduces environmental and economic costs, although high-temperature processing limits the scalability of such systems (Huang et al., 2023). Chemical methods for the production of silicon nanostructures from end-of-life silicon solar cells are also relatively costly (Deng et al., 2021). Thus, other applications, such as construction, are more economically and environmentally sustainable (Wolf and Stammer, 2024).”
Reviewer:
7. The tables summarize several photocatalytic systems but provide limited information on test conditions and pollutants. At present, they are dominated by dye degradation (methylene blue, rhodamine B, methyl orange). To broaden relevance, examples involving pharmaceuticals, pesticides or other emerging contaminants should be included where available.
Answers:
A few reports about the possible application of silicon-based nanosystems for real-life contaminants were discussed in the revised manuscript:
“Besides standard compounds used for testing the feasibility of pollutants’ degradation, several silicon-based systems were used for real-world contaminants. Sulfur- and nitrogen-doped silicon was used for efficient degradation of tetracycline (Wang et al., 2022). Successful employment of silica-TiO2 composites for photo-degradation of water-soluble pharmaceuticals (Gusmao et al., 2022) and pesticides (Kalidhasan and Lee, 2022, AbuKhadra et al., 2020), demonstrates the potential of the oxidized surface of Si-nanosystems for this purpose.”
Reviewer:
8. Most referenced studies rely on UV irradiation. Since UV-driven photocatalysis is less practical for real-world application, more emphasis should be placed on visible-light or solar-driven systems. The review should systematically highlight which Si-based systems are active under visible light and compare their potential for scale-up.
Answer:
We addressed this point in the revised manuscript:
“By the way, a larger number of examples reported in Table 1 demonstrate efficiency under real or simulated sunlight. This makes silicon-oxide interfaces promising candidates for real-world applications despite the performance inferior to more sophisticad materials, which will be discused below.”
“Table 2 contains a summary of the recent advantages of photocatalytic NWRs based on Si in organic pollutants removal processes. As one can see, the best performance of these materials corresponds with UV irradiation. However, the results obtained for materials demonstrates potential of Si-based NWRs for photdegradation of organic compounds.”
We also pointed out that important issue in conclusion:
“”
Reviewer:
9. Section 5. This section is useful but remains descriptive. The authors should expand the discussion by analyzing how ML could be applied not only to predict redox potentials or material stability but also to accelerate high-throughput synthesis and screening of Si-based photocatalysts. The review should also point out the limitations (e.g., scarcity of large, high-quality datasets) and the need for open databases in this field.
Answer:
In the revised manuscript, we clarified the current successes of ML:
“Thus, ML is an important tool to help in the area of interpretation of raw data.
…
In summary, these methods successfully solved the limitation of DFT-based methods in the number of atoms, which is essential for the simulations of realistic nanosystems and surfaces with complex morphologies.”
And pointed out the crucial step for the future development in this area:
“The key difficulty with building datasets suitable for ML requires the acceptance of a standardized description of experimental conditions and results. In current situations, the key experimental data for this type of dataset is extracted manually, which limits the size of these datasets.”
Reviewer:
10. The conclusion section is currently a summary of previous sections. It should instead articulate: (i) key knowledge gaps (e.g., photocorrosion resistance, scalability, dataset limitations for ML), (ii) challenges for future research (e.g., stability under natural sunlight, testing with real wastewater, integration into hybrid treatment systems), and (iii) opportunities for new research avenues (e.g., machine learning-guided design, valorization of industrial waste streams for Si precursors, coupling photocatalysis with biological treatments).
Answer:
Thank you, the conclusion was reordered with significant additions related with the most prospective (as we see it) directions for the future development in the area.
Reviewer:
Minor comments
- The abstract is generally clear but could better highlight novelty compared to prior reviews on Si-based photocatalysts.
Answer:
The abstract was partially rewritten.
Reviewer:
- Several figures (e.g., Figures 1–3) appear to be re-adapted from other sources. The figure captions should explicitly state permissions/licenses where necessary.
Answer:
The copyrights were added to the captions.
Reviewer:
- Some references are outdated; inclusion of more recent 2023–2025 works on visible-light photocatalysis and wastewater applications would improve the review.
Answer:
Multiple citations for the recent works were added in the revised manuscript.
Reviewer:
- English grammar and sentence structure are acceptable overall but require polishing in some sections for conciseness and clarity.
Answer:
Multiple sentences were rewritten to improve clarity.
Round 2
Reviewer 1 Report
Comments and Suggestions for Authors
Manuscript can be accepted now.
Reviewer 2 Report
Comments and Suggestions for Authors
Autjors have addressed the comments and questions presented and thus the manuscript can be published in the current form.